

# Regionalization of GR4J model parameters for river flow prediction in Paraná, Brazil

Louise Akemi Kuana [1], Arlan Scortegagna Almeida [2], Emílio Graciliano Ferreira Mercuri [3], and Steffen Manfred Noe [4]

[1]Programa de Pós-Graduação em Engenharia Ambiental, Universidade Federal do Paraná, Curitiba, Brasil.
[2]Sistema de Tecnologia e Monitoramento Ambiental do Paraná (Simepar).
[3]Departamento de Engenharia Ambiental, Universidade Federal do Paraná, Curitiba, Brasil.
[4]Institute of Forestry and Engineering, Estonian University of Life Sciences, Tartu, Estonia.

**Correspondence:** E.G.F. Mercuri (emilio@ufpr.br)

**Abstract.** Regionalization methods dependent on hydrological models comprise techniques for transferring calibrated parameters in instrumented watersheds (donor basins) to non-instrumented watersheds (target basins). This study aims to evaluate regionalization methods for transferring GR4J parameters and predict river flow in catchments from the south of Brazil. We created a dataset for Paraná state with daily hydrological time series (precipitation, evapotranspiration, and river flow) and watershed physiographic and climatological indices for 126 catchments. Rigorous quality control techniques were applied to recover the rainfall history from 1979 to 2020, and manual efforts were made to georeference the fluviometric stations. The regionalization methods compared in this study are based on: simple spatial proximity, physiographic-climatic similarity and regression by Random Forest. Direct regression of $Q_{95}$ was calculated using Random Forest and compared with indirect methods, i.e. using regionalization of GR4J parameters. A set of 100 basins were used to train the regionalization models and another 26 catchments, pseudo non-instrumented, were used to evaluate and compare the performance of regionalizations. The GR4J model showed acceptable performances for the sample of 126 catchments, 65% of watersheds presented log-transformed Nash-Sutcliffe coefficient greater than 0.70 during validation period. According to evaluation carried out for the sample of 26 basins, regionalization based on physiographic-climatic similarity showed to be the most robust method for prediction of daily and $Q_{95}$ reference flow in basins from Paraná state. When increasing the number of donor basins, the method based on spatial proximity has comparable performance to the method based on physiographic-climatic similarity. Based on the physiographic-climatic characteristics of the basins, it was possible to classify 6 distinct groups of watersheds in Paraná. The basins showed similarities in their size, forest cover, urban area, number of days with more than 150 mm of precipitation, and average duration of consecutive dry days.

## 1 Introduction

The hydrological model, GR4J (*Génie Rural à 4 paramètres Journalier*), proposed by Perrin et al. (2003) has been implemented in different countries, such as France (Oudin et al., 2008, 2010), Australia (Pagano et al., 2010), Brazil (Neto et al., 2021), South Korea (Shin and Kim, 2016), Mexico (Arsenault et al., 2019), and Russia (Ayzel et al., 2019), that is, in regions with varied



climatic, geological, pedological, and vegetation cover characteristics. This parsimonious model has been showing promising results and stands out for its dependence on few parameters and the use of two meteorological forcing variables on a daily scale, which are: the total precipitation and potential evapotranspiration averaged at the basin scale, requiring historical series of observed flows to adjustment of its four parameters. In view of recurrent absence of data in the field of hydrology, the International Association of Hydrological Sciences (IAHS) spread between 2003 and 2012 the movement that became known as the PUB decade (Predictions in Ungauged Basins) (Sivapalan, 2003).

According to Razavi and Coulibaly (2013), regionalization methods dependent on rainfall-runoff models comprise techniques for transferring calibrated parameters in instrumented basins (donor basins) to non-instrumented basins (target basins). The study carried out by Arsenault et al. (2019) presents three techniques based on physical similarity, spatial proximity and regression to estimate parameters of three different hydrological models, with the purpose of predicting continuous flows in watersheds that do not have monitoring. This study was carried out in Mexico, a region that can be considered heterogeneous in multiple points of view (e.g. climatology and pedology), and points out that these regionalization methods show to be more promising in humid regions when compared to arid regions. Although many advances have already been made in this area of hydrology, even 10 years after the conclusion of the PUB decade, there are still uncertainties arising from the structures of models, methods and procedures for estimating flows in ungauged basins (Guo et al., 2020). Part of this is due to the uniqueness of each region on the globe, which concerns not only the uniqueness of each location, but also the issue of availability of information (e.g. descriptive characteristics of basins and availability of hydrometeorological data). Additionally, there is the complexity of hydrological systems, which can be seen as the result of different processes that depend on temporal and spatial scale and interactions between climate, vegetation, topography, and soil (Blöschl et al., 2013; Hrachowitz et al., 2013), that make the task of estimating hydrological information in basins with little or no information challenging.

The study area chosen for this work is the State of Paraná, located in the southern region of Brazil with area of approximately $199,315 \text{ km}^2$. The hydrography of Paraná is composed mainly by Iguaçu River, Paraná River, Paranapanema River, Tibagi River, Ivaí River, and Piquiri River, which supply water to population and industries, generate energy, and irrigate crops (Carneiro et al., 2020). The State of Paraná faced one of its worst droughts in its history in 2020 and 2021 (Juliani et al., 2020). As a result, available hydrometeorological information is key to understand and plan an economy dependent on water supplies.

The aim of this article is to improve the methodology for transferring parameters of the GR4J model calibrated in instrumented watersheds to predict daily flows in basins with little or no hydrological information. The performance of the different methods implemented here is verified in Paraná basins that have a history of hydrometeorological data records. Other objectives are the following: i) build a hydrological database for Paraná State, Brazil. The database consists of daily flow, precipitation and evapotranspiration time series and catchments descriptive indices; ii) employ methods for transferring GR4J calibrated parameters through regionalization techniques based on spatial distance, physiographic-climatic similarity and non-linear regression; and iii) compare regionalization methods and Random Forest to estimate the $Q_{95}$ reference flow.





## 2   PUBs methods and regionalization


PUBs (Predictions in Ungauged Basins) methods arose from the need to reliably predict hydrological variables in basins that have little or no hydrometeorological information. According to Hrachowitz et al. (2013), such methods propose to extract as much information as possible from the data available in a region and, in cases where there is not enough data, they intend to investigate, explore and evaluate the applicability of new means to acquire this information. GRACE mission (Gravity

Recovery and Climate Experiment) is one example of new technology for groundwater observation, in which two spacecraft are used to detect the variation of gravitational field over time in a given region, and from these changes scientists are able to estimate the water table level variation (NASA, 2021). Regionalization methods aim to transfer hydrological information from one or more instrumentalized river basins to other non-instrumented catchments (Oudin et al., 2010; Guo et al., 2020; Blöschl et al., 2013). According to Razavi and Coulibaly (2013), the prediction of continuous flows in non-instrumented basins can be

performed using hydrological models with a physical, conceptual or empirical basis. Hydrological models calibration is carried out using observed flow data and, therefore, constants from one basin can be transferred or estimated in basins that do not have measurements.

Watersheds are complex systems that can be seen as the result of different processes, which depend on the time scale and interactions between climate, vegetation, topography and soil (Blöschl et al., 2013). Intuitively, the choice of receiving

(non-instrumentalized) and donor (instrumentalized) basin pairs can consider spatial proximity and/or physiographic-climatic similarity (Oudin et al., 2008, 2010; He et al., 2011; Razavi and Coulibaly, 2013; Guo et al., 2020). Therefore, both are part of distance-based regionalization techniques (He et al., 2011).

The application of the physiographic-climatic similarity method implicitly considers two assumptions. The first assumption is that if there is similarity between basins, there are similar hydrological responses. The second assumption is that the similarity

between sets of calibrated parameters of the hydrological model between two or more river basins may reflect on the similarity of their behavior in relation to the transformation of rainfall into flow (Oudin et al., 2008, 2010; Parajka et al., 2005; Blöschl et al., 2013).

Spatial proximity assumes that neighboring basins have similarities in climate, soil type, land use and cover, slope, altitude and other characteristics (Arsenault et al., 2019). The study carried out by Kuentz et al. (2017), which used more than $35,000$

basins and $1,366$ fluviometric stations in Europe to explore the correlation between 16 different indices of hydrological behavior responses (e.g. baseline flow index, $Q_5$ and $Q_{95}$) and 35 physical descriptors (e.g. area, slope and aridity index), concludes that there are strong connections between the physical descriptors and the response rates of the hydrological behavior.

According to He et al. (2011), spatial proximity regionalization techniques can be subdivided into: simple proximity and spatial interpolation. Simple proximity identifies the closest basins, and is often calculated using Euclidean or Haversine distance,

depending on the projection used. Parameters can be transferred in full or estimated through averages between basins within a certain radius (Guo et al., 2020). Among the spatial interpolation methods, kriging is one of the most popular techniques according to Razavi and Coulibaly (2013), where parameters are estimated through the weighted average of known values in the neighbourhood.





Another well-known category of regionalization is based on regional regression analyses (He et al., 2011). Linear or non-linear multiple regression models seek to equate the relationship between dependent variable (e.g. hydrological model parameters) with different independent variables (e.g. descriptive characteristics). Mohamed et al. (2019) explain that due to non-linear and multidimensional relationship between basins descriptive characteristics and model parameters, the application of regression methods that use machine learning techniques (such as Random Forest) are becoming more common to extrapolate hydrological models parameters.

More recently, according to Guo et al. (2020), there is a new classification called hydrological signature methods. Response indices of the hydrological behavior (e.g. permanence curve) estimated for non-instrumented basins are used as a reference for identifying model parameters that best reproduce these indices. Pinheiro and Naghettini (2013) showed acceptable performances for prediction of continuous flow by calibrating RIO GRANDE model using the synthetic permanence curves of Pará and Paraopeba river basins. On the other hand, Kim et al. (2017) when using regionalized permanence curves to calibrate GR4J model in a set of 45 South Korean basins, observed a decrease in the predictive capacity. Kaviski et al. (2002) developed a regionalization study in basins from Paraná, south of São Paulo and north of Santa Catarina States, for determining average, minimum and maximum flows. Bazzo and Almeida (2016) used Artificial Neural Network (ANN) algorithms to regionalize parameters of a rainfall-runoff model and thus estimate continuous flows in 15 watersheds in the interior of Paraná based on time series of rainfall, temperature and river flow.

## 3 Method

First part of this research was the data set construction for Paraná State and some surrounding basins. It consists of precipitation, evapotranspiration, and river flow data; also a set of descriptors for each catchment was calculated, they are based on: localization, landscape, relief, climate, hydro topography, land use/cover and soil information. After the dataset definition, calibration and validation of the GR4J hydrological model was performed in all basins. Three regionalization methods of the GR4J parameters were tested, they are based on i) physiographic-climatic similarity, ii) simple spatial proximity and iii) non-linear regression. Estimates of $Q_{95}$ flow using a machine learning algorithm based on the dataset was also compared with data. All these methods are explained in next sections.

### 3.1 Dataset for Paraná State

The study area was delimited based on the hydrographic network of Paraná State, Brazil, which is available at Instituto Água e Terra (IAT) (IAT, 2020), and on a rectangular polygon demarcated between latitudes 22° 15' 36"S and 26° 54' 00"S, and longitudes 48° 00' 00"W and 54° 42' 00"W. Therefore, study area includes the Paraná State, and extends into parts of Santa Catarina and São Paulo States, not completely covering Paranapanema and Paraná river basins (Figure 1).



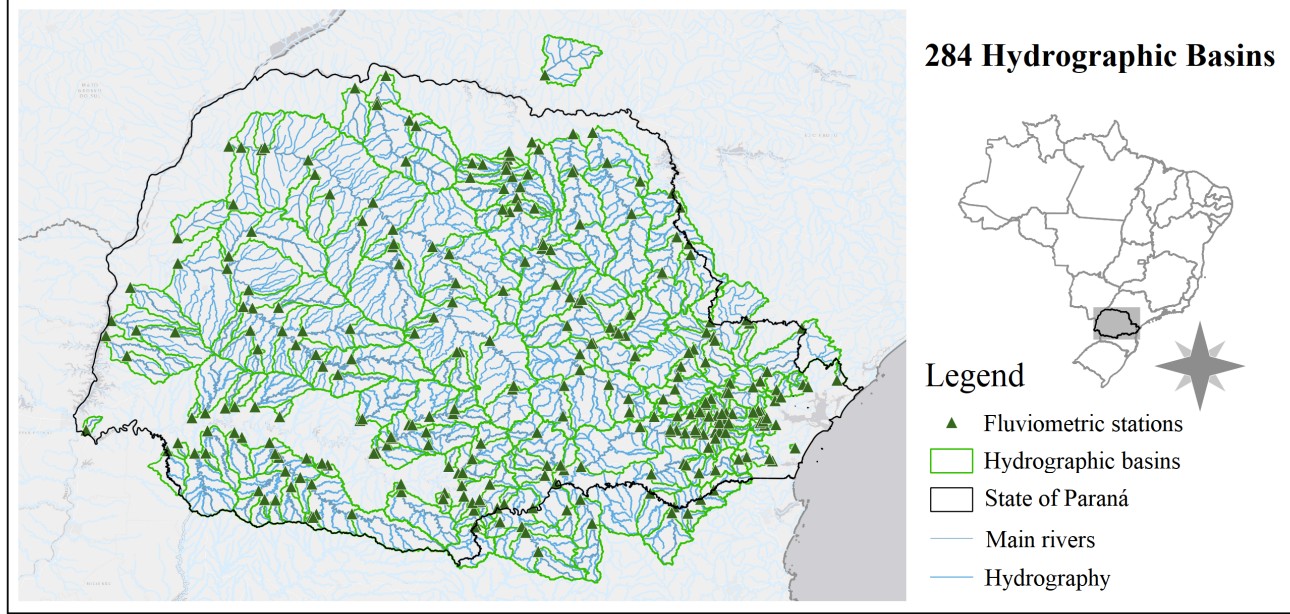

**Figure 1.** River flow stations and watershed delineation.

Time series of hydrometeorological observations were obtained at: Agência Nacional de Águas e Saneamento Básico (ANA) via HidroWEB Portal, Instituto Nacional de Meteorologia (INMET), Sistema de Tecnologia e Monitoramento Ambiental do Paraná (Simepar), Água e Terra Institute (IAT) and Instituto de Desenvolvimento Rural do Paraná (IAPAR-EMATER).

Although there are datasets at national level, such as CAMELS-BR (Chagas et al., 2020) and CABra (Almagro et al., 2021), the authors decided to construct a new dataset based on the hydrographic network of Paraná State. This network has a consistent topology and codification, it's hierarchization was proposed by Otto Pfafstetter (Pfafstetter, 1989) and allows extraction of information upstream and downstream of each river section (Sousa et al., 2009).

### 3.1.1 Precipitation time series

The Llabrés-Brustenga et al. (2019) quality control method was used to evaluate daily rainfall data series from 1389 stations, which are shown in Figure 2. The method can be divided into four steps. First, stations coordinates, the period of operation and percentage of available data are checked. In second stage, data that are not physically possible are identified and discarded, such as: negative precipitation values and extreme events greater than 300 mm.



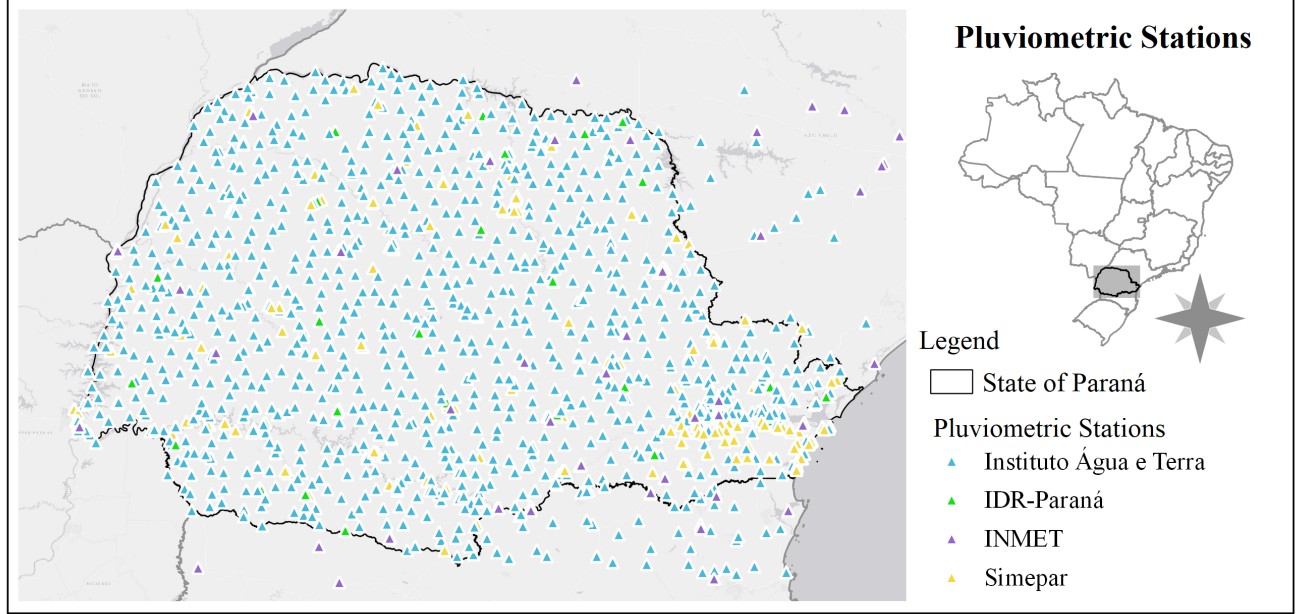

**Figure 2.** Location of pluviometric stations.

Third stage consists of analyzing the historical series of each station individually. For each year, a quality index is calculated, which depends on 5 (five) factors, namely: i) Percentage of data available in each year of the series; ii) Distribution of failures throughout the year. For this factor, the penalty becomes greater for a series that had long continuous periods of failures; iii) Probability that the series is formed by possible failures that have been padded with zeros. This factor penalizes the series that had monthly cumulative data equal to zero, indicating that "false zeros" are possible; iv) Probability of systematic accumulation of two or more days of the week. This factor penalizes the station if a day of the week with a tendency to rain less than other days of the week is detected at the station; and v) Probability that the series contains outliers. The quality index can range from 0 (zero) to 100 (one hundred) percent. Values equal to 100% indicate absolute quality, above 80% is considered acceptable and below 50% the quality is very low.

The verification process between what was recorded at the station to be analyzed (candidate station) and at neighboring stations (auxiliary stations) is carried out in the fourth stage, also known as relative quality control. At this stage, there are two indices that are relevant to the classification of daily values for the candidate post, which can be labeled as: valid, "V", doubtful, "D", invalid, "N", or insufficient information, "I". The records identified as insufficient information denote that there are less than two auxiliary stations in the region and in the same period to be properly evaluated. The first index, called representativeness index, verifies the daily values for each station pair, candidate - auxiliary, this index considers the distance between stations, the altitude difference and the correlation with measured data. A maximum distance of up to 50 km was defined between candidate and auxiliaries stations. The second index is used to analyze the monthly cumulative data. From



the Simepar stations, maximum limits of monthly accumulated were established, which were applied to evaluate the historical series of each candidate station.

Validated daily data were spatialized on a 1 km × 1 km grid using space-time kriging, where precipitation values were estimated on a daily scale using weighted averages between neighborhood data. Then, a second spatialization method was applied to the most recent precipitation history. The method presented by Calvetti et al. (2017) was used to estimate precipitation spatially within the area of interest, where the Poisson equation was used to combine radar and satellite data with the records observed by telemetry precipitation stations. Finally, average rainfall was measured using the arithmetic mean of the grid points located within the drainage area of each watershed.

### 3.1.2    River flow data and delineation of watersheds

We constructed the river flow dataset in two ways: i) by directly obtaining river flow time series, where observed water levels were previously transformed into flow by the agency responsible for operating the station, and ii) by time series of water levels which still had not been transformed into flow and, therefore, when available, the station's rating curves were obtained and then the quota was transformed into flow. We have considered acceptable to use stations with at least 5 years of river flow records for the application of regionalization methods. Time series from stations downstream of flow regularization (dams and reservoirs) were discarded or considered partially in the periods prior to the dams construction.

Conventional stations were obtained from the IAT and HidroWEB Portal databases. Although both banks preserve information from stations of different operators, it was adopted that information coming from the IAT bank would have priority over the ones from HidroWEB Portal. The inventory provided by the technician responsible for the IAT informs that there are 413 river flow stations in the study area with time series bigger than 5 years. From the ANA metadata catalogue, only 15 different stations with at least 5 years of river flow records were identified. The telemetric series of 83 IAT stations and 57 Simepar stations were obtained from the Simepar database.

Locations of the stations were checked through the manual procedure of hydroreferencing using the hydrographic network of the IAT. Finally, a manual quality control was carried out, in which non-consistent data were disregarded, such as: sudden ruler changes clearly altering the base flow, series with large gaps alternating with short measurement periods, low precision measurements or that presented constant values for long periods. In the end, a total of 284 river flow stations were obtained with observations ranging from 1926 to 2020, as shown on the map in Figure 1.

### 3.1.3    Potential evapotranspiration

The FAO Penman-Monteith (Allen et al., 1998) equation was used to estimate potential evapotranspiration (ET). This method requires time series of air temperature, air relative humidity, wind speed and solar radiation, which were obtained through Simepar telemetric stations with records ranging from 1997 to 2020. ET was based on long-term average daily values, which means the same potential evapotranspiration series was repeated every year for each station. Subsequently, the punctual information was spatialized using a method of regression followed by interpolation, also known as Regression-Kriging or hybrid method of interpolation (Hengl et al., 2007).





## 3.2 Catchment Descriptors

Table A1 shows catchment descriptors statistics (mean, standard deviation, quartiles, minimum and maximum) for the 126 basins of Paraná dataset. It has 39 descriptive indices divided in 4 categories: physiographic, climatological, land use / land cover, and soil type. Quantitative indices were used to describe landscape, relief, climate, topology of the hydrography, land use and soil type of the watershed. Physiographic indices were obtained for each geographic location and topography of the drainage networks for the selected basins. From the hydrographic network, areas and drainage sections were obtained, which were used as a basis for calculating the indices described in Table A1, from the appendix. Digital elevation model (DEM) with a resolution of 30 meters from NASA's Shuttle Radar Topography Mission (SRTM) was used to estimate slopes and altitudes. Land use and land cover maps for the year 2019, provided by MapBiomas (Souza et al., 2020), were used for calculating the fractions of area that each class occupies in the basins and to determine the dominant class. Soil map was obtained from Embrapa (2020) for Paraná State with a scale of 1:250,000.

The curve number method (CN) developed by Soil Conservation Service (1972) relates soil and land use and land cover information to classify the region based on its storm water retention potential. The ANA metadata catalog was used to estimate the CN in Paraná basins. Average precipitation series and potential evapotranspiration estimates, which were previously determined for each watershed, were used to calculate the indices related to precipitation and potential evapotranspiration. Furthermore, Barbieri et al. (2017) provided atlases of Paraná State with monthly average temperatures and average solar radiation for each season of the year. The atlases were produced based on measurements from INMET, Simepar and Instituto de Desenvolvimento Rural do Paraná (IDR-Paraná) stations between the period 2006 to 2016. This information was used to compute average indices in basins located within the state, and for the catchments on borders or among other states, average values of the nearest watershed were adopted.

## 3.3 Watershed Selection

The watershed selection consists of 126 river basins that have at least 15 years of flow data between 1979 and 2020, and each year counted has a maximum of 10% of gaps. In addition, it was preferred that the historical series also had more recent data, which extended beyond the year 2010, and limited to homogeneous historical series that passed the Pettitt test (Pettitt, 1979). The non-parametric Pettitt test was calculated using the library *Pyhomogeneity* in *Python*, it is able to indicate the year in which a sudden change in the temporal trend occurred. The selected 126 watersheds are depicted in Figure 3. Figure A1 (appendix) shows the availability of data over the years, darker green indicates greater amount of data available in that year.



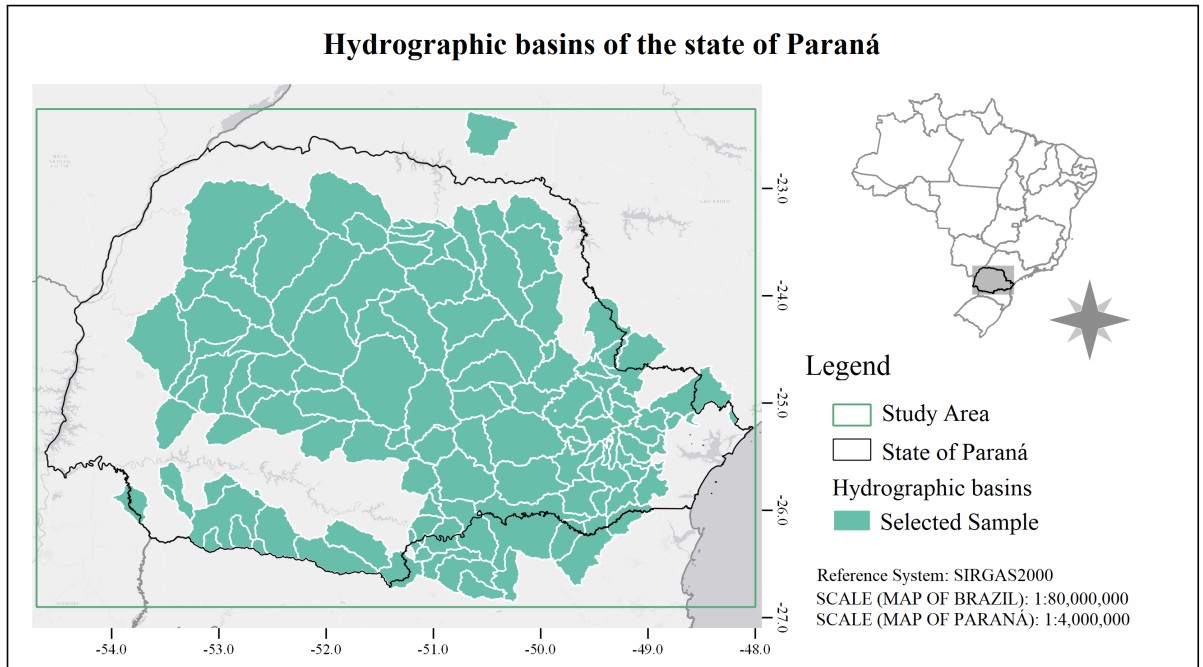

**Figure 3.** Location of selected watersheds in the state of Paraná.

## 3.4 Calibration and Validation of the hydrological model

GR4J model parameters are obtained through calibration, a process of making simulated flow as close as possible to observed flow. Table 1 summarizes minimum and maximum values used for searching each constant of the model.

**Table 1.** Descriptions and ranges for GR4J model parameters.

| Parameters | Description | Interval |
|:---:|---|---|
| $X_1$ | Production tank capacity (mm) | 0 to 6000 |
| $X_2$ | Coefficient of underground exchanges (mm/day) | -20 to 10 |
| $X_3$ | Propagation reservoir capacity (mm) | 0 to 4000 |
| $X_4$ | Unit Hydrograph Base Time (days) | 0.04 to 20 |

Simulation period was divided into 3 parts: warm-up, calibration and validation. First 5 years of simulation were used as a warm-up to eliminate the uncertainties of initial conditions (Daggupati et al., 2015). Calibration and validation periods were defined as 70% and 30% of the remaining time series after warm-up, respectively and sequentially.

*Differential Evolution* (DE) optimization method was used for GR4J calibration. This method was initially proposed by Storn and Price (1997) and is part of *Scipy* library in *Python*. DE is used in optimization problems that use a single objective function,



which is our case. According to Krause et al. (2005) and Muleta (2012), the use of Nash-Sutcliffe logarithmic coefficient
(logNSE) as an objective function is more influenced by low flows and, therefore, this metric can be used to evaluate the
performance of minimum flows predictions. The logNSE can range from $-\infty$ (poor fit) to $1.0$ (perfect fit), and is calculated
as:

$$
\mathrm{logNSE} = \frac{\sum\limits_{i=1}^{N} (\ln(Q_i^{\mathrm{sim}} + 0.001) - \ln(Q_i^{\mathrm{obs}} + 0.001))^2}{\sum\limits_{i=1}^{N} (\ln(Q_i^{\mathrm{obs}} + 0.001) - \bar{Q}_{\mathrm{ln}}^{\mathrm{obs}})^2}, \tag{1}
$$

where $Q_i^{\mathrm{sim}}$ and $Q_i^{\mathrm{obs}}$ correspond to simulated and observed flow on day $i$, respectively. The average term $\bar{Q}_{\mathrm{ln}}^{\mathrm{obs}}$ is calculated
by $\bar{Q}_{\mathrm{ln}}^{\mathrm{obs}} = \frac{1}{N} \sum_{i=1}^{N} \ln(Q_i^{\mathrm{obs}} + 0.001)$.

Other metrics used to evaluate the performance of regionalization methods are the Pearson Correlation Coefficient (R), the
Nash-Sutcliffe Coefficient (NSE) and the Nash-Sutcliffe square root coefficient (sqrtNSE) where flow is transformed by the
square root.

## 4   Regionalization Methods

In this work, classical regionalization techniques based on physiographic-climatic similarity, simple spatial proximity and non-
linear regression were used. Regionalization based on physiographic-climatic similarity starts by identifying and grouping the
watersheds that have the greatest physical, climatic and geographic similarities. The purpose of clustering is to identify ho-
mogeneous regions based on descriptive indexes. Regionalization based on simple spatial proximity considers that the study

region is homogeneous and, therefore, nearby basins are similar based on climate, relief, vegetation, landscape and soil type.
Although both assume that physical similarities can be closely correlated with hydrological responses, if the region is heteroge-
neous, regionalization based on physiographic-climatic similarity transfers information between basins that are not necessarily
geographically neighbours. A second assumption to be considered is: the similarity between a a parameters from two or more
river basins may reflect on the similarity of their behaviour in relation to the transformation of rainfall into flow (Oudin et al.,

2008, 2010; Parajka et al., 2005; Blöschl et al., 2013). On the other hand, regression methods consider that hydrological model
parameters may be related to some physical processes that occur in watersheds and, consequently, are associated with some
descriptive characteristics (Arsenault et al., 2019). In this way, it is possible to build a regression model for each parameter of
the model.

Diagram in Figure 4 briefly summarizes the application of regionalization methods in this work. After calibrating the GR4J

model for each of the 126 river basins, catchments were randomly divided into training and validation sets, with 80% of initial
sample basins comprising the training set and 20% forming the validation set.

The training set is formed by river basins considered as possible donors of GR4J parameters, and which were also used to
train and build regionalization models. Basins of the validation set are considered to be pseudo non-instrumentalized (indicated
by the blue arrow in the diagram of Figure 4), even if it is known that these catchments have hydro meteorological data and





were calibrated. Each of regionalization methods consists basically in different methodologies for selecting donor basins for transferring the parameters of GR4J model to target basins.

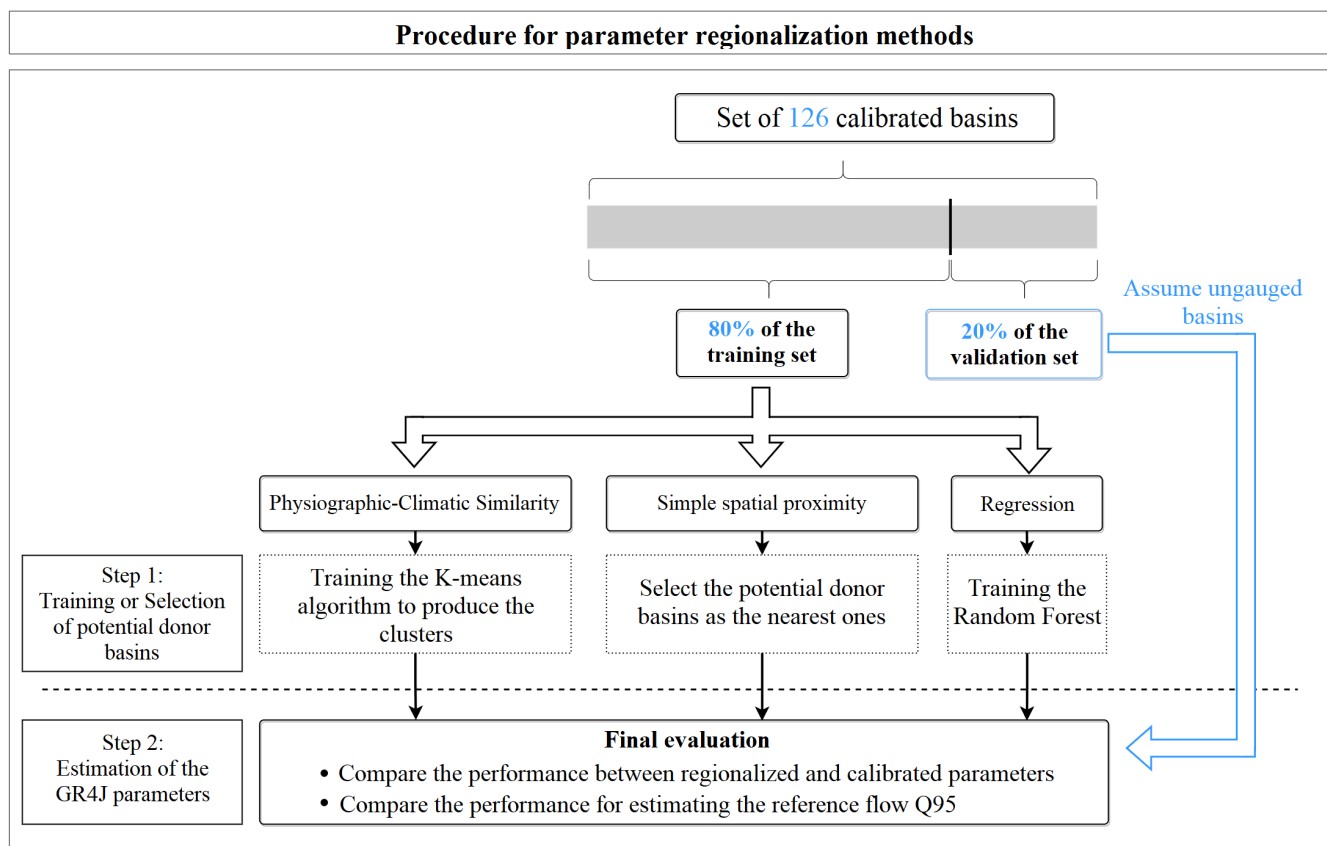

**Figure 4.** Diagram summarizing the regionalization methods.

## 4.1    Physiographic-Climatic Similarity

When applying PUBs methods we must take into account the uniqueness of each region on the globe and all information available in each data set. Bearing in mind the uniqueness of each location, possible basin descriptors were carefully chosen
so that they would synthesize different characteristics of river basins and be capable of transmitting the diversity between catchments within the same sample. Thus, the following descriptors were initially selected: basin area, length of main river, altitude and average basin slope, latitude of the basin centroid, daily averages of precipitation and potential evapotranspiration, aridity index, average number of days with extreme precipitation events and fraction of area covered by forest, agriculture and urbanization.
255        Descriptors were normalized so that mean and standard deviation corresponded to zero and one, respectively. This procedure ensures that different variables share the same scale without significant loss of information and, thus, allows categories with





different magnitudes to be compared equally. Then, characteristics that showed variability were selected, i.e., that described the set of watersheds as being heterogeneous.

Table A1 shows the descriptive statistics adopted, it reveals that Paraná basins have diverse area, length of main river
and average altitude. On the other hand, the fraction of urban area showed little variation; despite this, it was preferable to keep this descriptor, since urban infrastructure as well as other anthropogenic activities can seriously disturb the processes of hydrological cycle.

High multicollinearity between the descriptors can lead clustering algorithms to take wrong decisions during the formation of groups (Boutsidis et al., 2014). Therefore, two analyses were performed to identify the correlation between descriptors. First,
Pearson correlation (R) between each pair of descriptors were calculated. Second, Variance Inflation Factor (VIF) was determined to measure the degree of multicollinearity between descriptors. IVF ranges from 1 (when there is no multicollinearity) to infinity (when there is perfect multicollinearity), the threshold used in this work was below 5. Correlations between descriptors can be seen in Figure B1 of Appendix B. High correlations, with R values above 0.70, were found between the following pairs of descriptors: aridity index and days of monthly accumulated precipitation above 150mm, average duration of days without
rain and latitude of basin centroid, average slope of the basin and fraction of forest, fraction of agricultural area and fraction of forest, annual potential evapotranspiration and average altitude of the basin. To reduce the dimensionality of data the following descriptors were selected: area, forest fraction, urban area fraction, average duration of extreme events with high precipitation (days of monthly accumulated precipitation above 150mm) and average duration of days without rain.

The Euclidean distance (dist), equation 2, is a metric that can express similarities (small distances) or differences (large
distances) between $n$ attributes of two basins ($a$ and $b$) in an n-dimensional space of attributes (Viviroli et al., 2009).

$$\text{dist}(a,b) = \sqrt{\sum_{k=1}^{n} \left[\text{atrib}_k(a) - \text{atrib}_k(b)\right]^2} \qquad (2)$$

Clusters were produced using the K-means method, which was implemented using the *scikit-learn* package. The application of K-means algorithm involves: first, define the number of $K$ groups, second, for each group initialize a centroid randomly within the range of each category, third, each point is assigned to the centroid that has the smallest distance Euclidean with
respect to point, four, a new location of $K$ centroids is computed based on the average of all points assigned to it. The iterative process from third to fourth step is repeated until there are no more changes in the centroids (Wilks, 2011).

The value of $K$ directly affects how groups will be formed. Increasing the number K leads to more groups, but consequently each group will have fewer members (which brings homogeneity, but does not guarantee representativeness). On the other hand, creating fewer groups generates groups with more members (which does not allow proper identification of the different
groups). Two ways to evaluate if the appropriate number of clusters resulting from the agglomeration method is using the Silhouette coefficient (Si) and the *Elbow* method.

According to Rousseeuw (1987), the Silhouette coefficient (Si) consists of calculating the average Euclidean distance ($a_p$) of a point $p$ with all points belonging to the same group. Then, the average distance ($b_p$) of the point $p$ with respect to all the





points belonging to the nearest neighbour group is calculated. Thus, the coefficient can be determined using the equation:

$$\text{Si} = \frac{b_p - a_p}{\max(b_p, a_p)}. \tag{3}$$

Si can vary between $[-1, 1]$, and the closer to 1 the more distant the point $p$ is from the neighbour group. Values close to 0 indicates that the point $p$ is close to the limit that divides both groups. And measurements close to $-1$ indicate that the point $p$ may have been associated with the wrong group.

The elbow method is a graphical tool for evaluating an optimal number of clusters. This technique involves calculating

an agglomeration coefficient, in this work the criterion used was the sum of squared distances of each sample ($x_i$) with the respective centroid ($\mu_j$) of the grouping that the sample is part of, which can be expressed as the sum of squared errors (SSE):

$$\text{SSE} = \sum_{i=1}^{n} (x_i - \mu_j)^2. \tag{4}$$

As number of clusters grows, distances between samples with their respective centroids decrease. However, the number of groups and the clustering coefficient are expected to be small. Thus, from a graph with agglomeration coefficient on $y$ axis

and number of groups on $x$ axis, it is possible to identify the point at which there is a sharp flattening or a rapid drop in this coefficient, suggesting an optimal number of clusters (Ketchen Junior and Shook, 1996).

### 4.2 Simple Spatial Proximity

The distance between two points, in this case the centroids of target and donor basins, that have known latitude and longitude can be calculated using the Haversine distance $D_H$:

$$D_H = 2r \arcsin\left[ \sqrt{\sin^2\left(\frac{x_1 - x_2}{2}\right) + \cos(x_1)\cos(x_2)\sin^2\left(\frac{y_1 - y_2}{2}\right)} \right], \tag{5}$$

where $D_H$ refers to the distance in km, $r$ is the average radius of the Earth, approximately 6371 km, $x$ and $y$ are respectively the latitudes and longitudes of points 1 and 2.

### 4.3 Regression

Multiple regression models, whether linear or non-linear, seek to find the best relationship between a dependent variable and

independent variables, this is done by finding the minimum error given a target. In our case the GR4J model parameters are dependent variables which will be calculated based on descriptive characteristics of the basins (independent variables). The non-linear regression method *Random Forest* (Breiman, 2001), which was chosen for this work, is able to perform well when dealing with large datasets and is able to distribute weights for the independent variables according to their degree of importance. Thus, two types of regression methods were constructed: *Random Forest I* and *Random Forest II*.

*Random Forest I* used a thousand decision trees and was trained using basin descriptors. For this technique, it is necessary to produce a regression model independently for each parameter ($X_1$, $X_2$, $X_3$ and $X_4$), however the parameters of a hydrological model generally present dependent relationships among themselves, and for sometimes cannot be observed independently.



Thus, a second method defined as *Random Forest (RF) II* included the calibrated parameters of training basins as descriptors. Second method followed the steps: i) a correlation analysis between GR4J parameters was performed, and thus an ordered list

of parameters from highest to lowest correlation index was created; and ii) first regression was done for the parameter with lowest correlation index, in this case only the descriptive characteristics were used to train the model. Then, regression was performed for parameter with second lowest correlation index, here we used descriptive characteristics and previous parameter that has the lowest correlation index. This process followed until all parameters had their regressions.

### 4.4   Q$_{95}$ Flow Estimate

Instituto Água e Terra (IAT), the environmental agency responsible for legal permissions of water resources in Paraná State, uses the river flow with 95% permanence (Q$_{95}$) as a reference flow rate for permission licenses of water use (AGUASPARANÁ, 2010). We have proposed to estimate Q$_{95}$ flow through regression techniques based on basin information and, thus, compare it with Q$_{95}$ flow calculated using regionalized simulated flows. The construction of permanence curves involved: (i) ordering the flows $Q$ in ascending order for $N$ days, (ii) assigning to each ordered flow $Q_m$ the corresponding ranking order $m$, (iii)

compute the frequency or probability of the ordered flows $Q_m$ to be equalled or surpassed $P(Q \geq Q_m)$, which can be calculated using the Weibull plot position shown in the following equation (Pugliese et al., 2014):

$$P(Q \geq Q_m) = 1 - \frac{m}{N+1}. \tag{6}$$

After obtaining Q$_{95}$ reference flows for the training set, a transformation of units from m$^3$/s to L/s/km$^2$ was performed, ensuring that the variable is not dependent on basin area. Then, another *Random Forest* regression method was trained and

evaluated for the test set. This RF used $1,000$ decision trees and watershed descriptors that presented weights greater than 0.01.

As terminology may sound ambiguous, it is important to distinguish here what are the training and test (or validation) sets used through this work. There are warm-up, calibration and validation periods for river flow simulation, and there are also training and validating sets for machine learning performance evaluation. The 126 basins were divided into training and validation sets for regionalization evaluation. Also, for estimating GR4J parameters and Q$_{95}$ reference flows another training

and test (or validation) sets were created for applying *Random Forest* regressions.

## 5   Results and Discussions

### 5.1   Performance of the GR4J model

GR4J model showed acceptable performances for the sample of 126 watersheds, as shown in Figure 5, with about 65% of Paraná watersheds presenting logNSE equal to or greater than 0.70 during validation period. Basins located close to Paraná

coastline reached a lower efficiency when compared to other regions. Some river basins presented superior performances in the validation period when compared to the calibration period, however inverse situations also occur. These phenomena may be associated with changes or improvements in measurement techniques, as well as influenced by changes in land use and land cover.





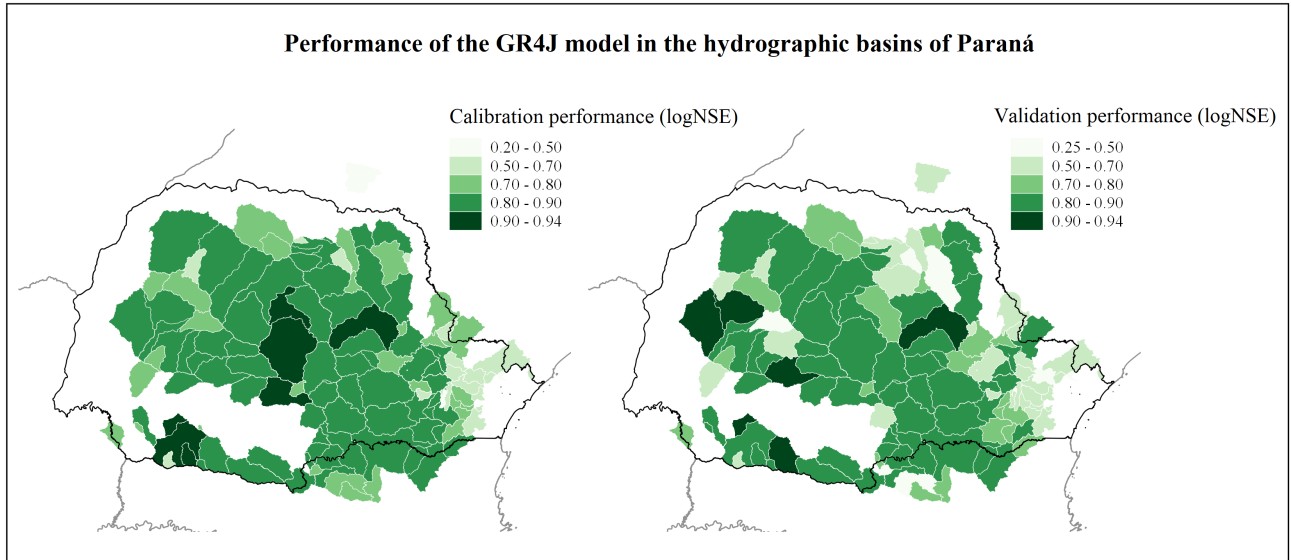

**Figure 5.** Performance of GR4J model during calibration and validation periods.

## 5.2 Performance of regionalization methods

Results of regionalization methods are described below one by one and, finally, the performance between them is compared.

### 5.2.1 Physiographic-Climatic Similarity

The regionalization method by physiographic-climatic similarity starts with defining the number $K$ of clusters used to group the basins. *Elbow* method indicated that $K = 6$ was appropriate, which is the point of abrupt slope change or curve flattening in Figure 6. Accordingly, Silhouette coefficient (Si) was higher when the number of clusters $K$ was equal to 6, as shown in

Figure 7. After defining $K$, we used 80% of 126 watersheds for training the K-means algorithm, which was used to group similar watersheds. The remaining 20% were used to test and evaluate the clusters formed.



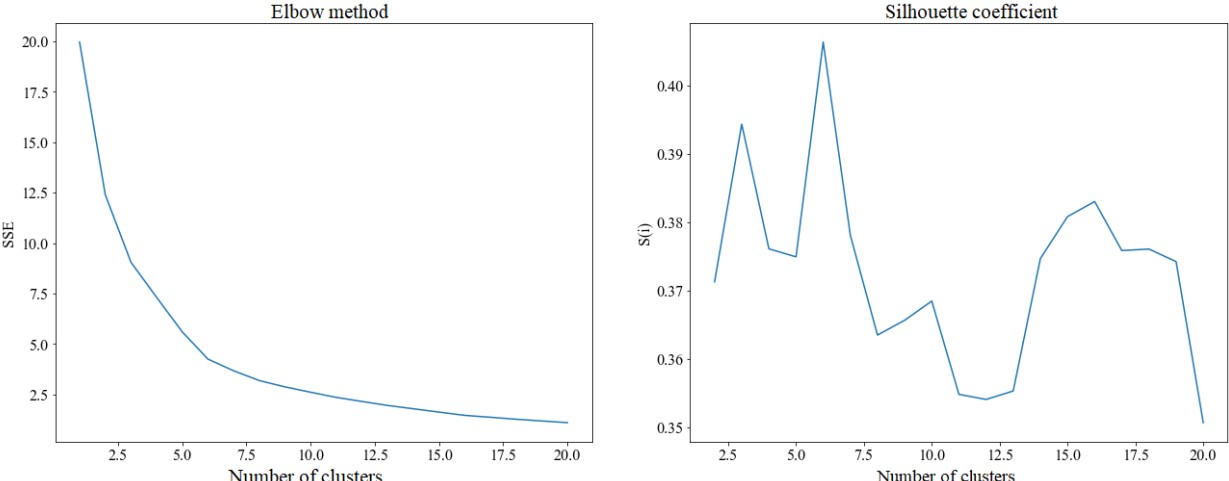

**Figure 6.** Elbow method for training set basins with $K$ ranging from 1 to 20.

**Figure 7.** Silhouette coefficients for training set basins with $K$ ranging from 1 to 20.

The geospatial distribution of watershed and clusters formed by K-means algorithm can be seen in Figure 8 for basins in training set (left) and validation set (right). Watersheds location in training set was similar to basins spatial distribution in validation set. Additionally, geographically close basins do not always belong to the same formed group.

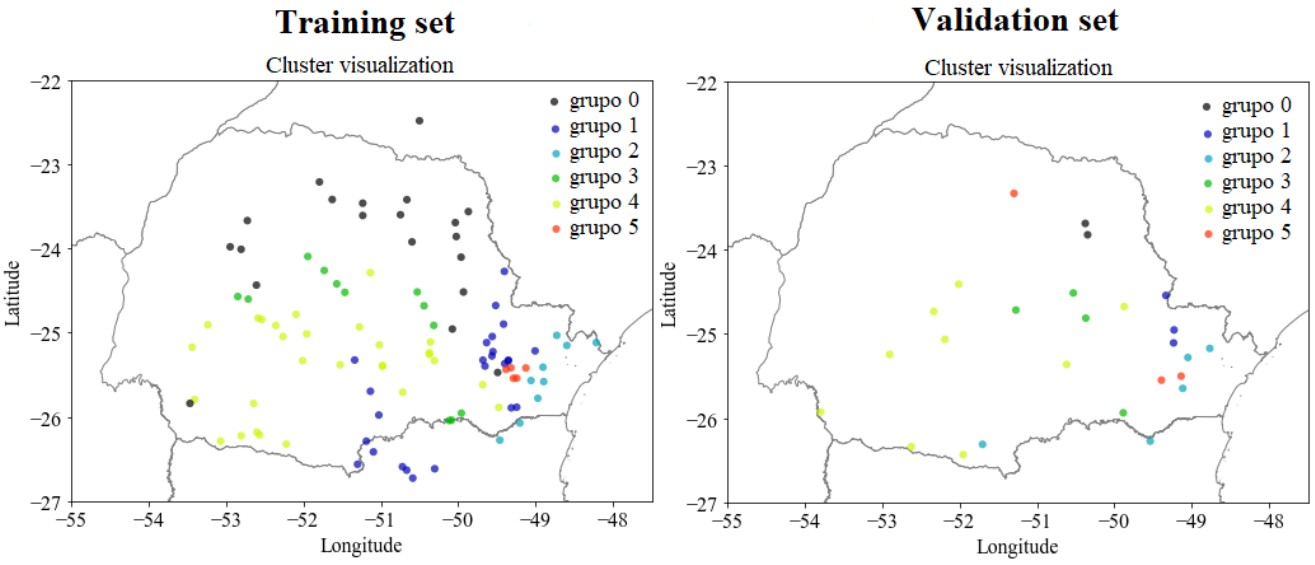

**Figure 8.** Clusters produced by the K-mean method for the 100 training basins (left side) and 26 validation basins (right side).

Descriptors distribution for each group in training set are shown in box-plots of Figure 9. Group 3 contains basins with the largest drainage areas, located in second and third plateaus of Paraná State centre. Group 4 contains basins that have





smaller drainage areas when compared to group 3, but which end up sharing similar characteristics to catchments in group 3. Groups 1 and 2 have a higher percentage of forest, but group 2 has a greater tendency to have more rainfall, smaller areas and shorter periods of consecutive dry days. On the other hand, group 0 stands out for containing the basins that have the longest average duration of consecutive dry days. Finally, group 5 stands out from the others because it contains basins with the highest percentages of urban area and, therefore, may be more influenced by anthropogenic activities. Although the descriptors point to heterogeneities between the formed groups, it is still possible to see overlaps, mainly in relation to the calibrated parameters of GR4J model, as shown in Figure 10.

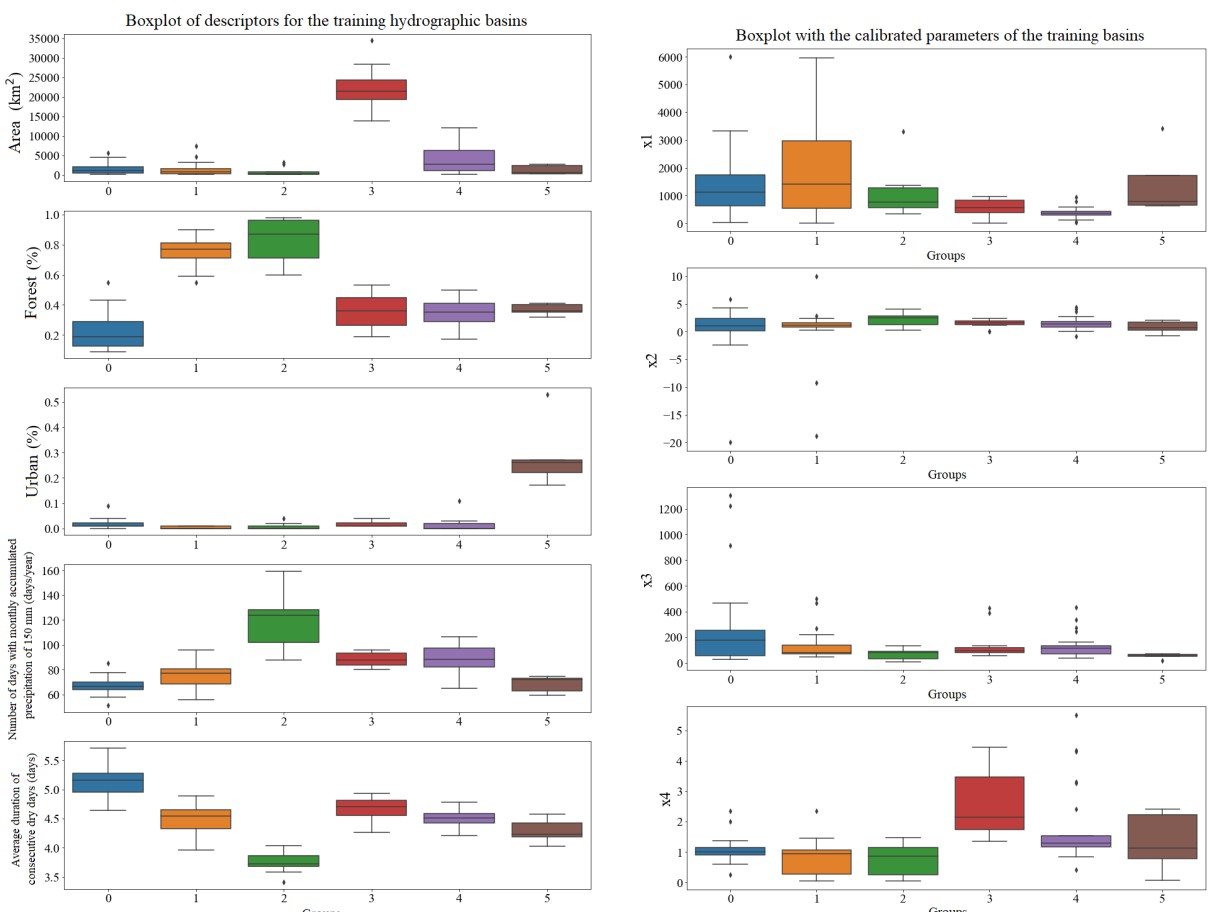

**Figure 9.** Basins descriptors distributions for training set clusters.

**Figure 10.** GR4J parameters distributions for basins in training set groups

Basins from group 1, which are located both on first and second plateaus of Paraná State, are similar in size to basins of group 0, but have a higher percentage of forest as a distinct characteristic. Basins of group 2 are found mainly in Paraná coastal region, near *Serra do Mar*, a long system of mountain ranges and escarpments, where orographic rain is more likely to occur. Group 4, which is present in greater quantity, contains hydrographic basins located in all plateaus of the State.



### 5.2.2 Simple Spatial Proximity

The simple spatial proximity regionalization method considers that the region near to the basin of interest is homogeneous
and, therefore, it has hydrological similarity. Assuming this hypothesis, we have used the Haversine distance between pairs
of receiving basins (pseudo-non-monitored) and donor basins (instrumented basins) to transfer of parameters from the GR4J
model. In both methods, physiographic-climatic similarity and simple spatial proximity, the receiving basins are all catchments
within the validation set, and the possible parameter donor basins are those from the training set that reached logNSE equal to
or greater than 0.70 during the validation period.

We have allowed more than one donor basin to transfer the GR4J parameters to target basins. When there is more than one
donor catchment, the four parameters of each donor basin was used to estimate flow in the pseudo non-instrumented target
basin. Once the flows were simulated with the donor basins parameters, the average of modelled flows were calculated and
used for the target basin.

We compared the ability of both methods (spatial proximity and physiographic-climatic similarity) to generate good results
in parameter regionalisation. For this we varied the number of donor catchments from 1 to 10 and evaluated the median logNSE
of the receiving catchments river flow simulations in the validation period. The analysis to identify the number of donor basins,
shown in Figure 11, indicates that the similarity method presented a maximum median logNSE for a total of 1 basin, and the
proximity method presented better results using 7 basins as donors.

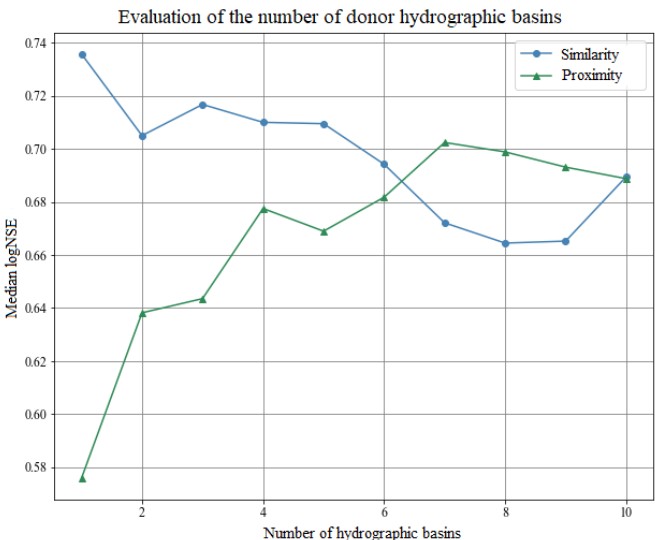

**Figure 11.** Evaluating the optimal number of donor basins based on logNSE medians during the validation period of receiving basins
simulations. The green and blue lines represent spatial proximity and physiographic-climatic similarity regionalization methods, respectively.





### 5.2.3 Regression of GR4J Parameters

To train the *Random Forest* regression model, known information about the watersheds in training set was used, namely: the descriptive characteristics (independent variables) and the calibrated parameters of the GR4J model (dependent variable). The descriptive characteristics that *Random Forest* pointed out as most relevant were: slope of main river and average slope of the basin, for parameter $X_1$; average altitude and average radiation in winter, for parameter $X_2$; average radiation in winter, for parameter $X_3$, and fraction of gleissol, for parameter $X_4$.

## 5.3 Comparing Regionalization Methods

In general, the vast majority of basins from validation group presented results with logNSE greater than 0.50 for different regionalization methods, and only two basins within this group presented low performance. Another general behavior was that basins with logNSE equal to or greater than 0.77 using calibrated parameters also achieved comparable performances with regionalized parameters. Additionally, in some cases where regionalization methods used more than one donor basin

they provided a diversified set of parameters. When combining this set of parameters they can result in superior performance compared to the use of calibrated parameters in a period prior to validation.

Results were evaluated using the Pearson Correlation Coefficient (R), the Nash-Sutcliffe Coefficient (NSE) and their variations: the flow transform by the square root (sqrtNSE) and by the logarithm (logNSE). NSE gives more emphasis on the performance of higher flows, logNSE is more sensitive to low flows and sqrtNSE provides an intermediate performance (Oudin

et al., 2008). Table 2 shows the median values of error statistics for estimated flows in validation period. Among regionalization methods, values that achieved the best results for each index are highlighted in bold. Thus, the physiographic-climatic similarity stands out positively by reaching logNSE and sqrtNSE equal to 0.736 and 0.726, respectively. Another point to be highlighted is that the spatial proximity method presents, in general, median results for the three coefficients. Table 2 also reveals that regionalization methods performances, in particular proximity and *Ranfom Forest*, can reach median NSE values

equal to or greater than when parameters were calibrated in a period prior to validation.

**Table 2.** Median values of error statistics calculated for validation period. In bold are shown the best results for each index among regionalization methods.

| | Efficiency metrics in the validation period | | | |
|---|---|---|---|---|
| | Calibrated | Proximity | Similarity | *Random Forest* |
| **NSE** | 0.621 | 0.635 | 0.602 | **0.643** |
| **logNSE** | 0.758 | 0.702 | **0.736** | 0.679 |
| **sqrtNSE** | 0.736 | 0.707 | **0.726** | 0.713 |

The review carried out by Guo et al. (2020) included the analysis of articles from different regions of the globe, which were recently published between 2013 and 2019, in which the researchers also applied similar regionalization techniques (proximity,





similarity and regression). Guo et al. (2020), based on these analyzed studies, show that there is evidence that regionalization methods based on distances (proximity and similarity) generally present superior performances to methods based on regression.

## 5.4 $Q_{95}$ Flow Estimation

In order to compare the performance of estimating the $Q_{95}$ reference flow between direct (regression) and indirect (regionalization of parameters) techniques. The regression method of *Random Forest*, which was named as *Random Forest* $Q_{95}$, was applied to directly regionalize $Q_{95}$ flow.

The construction of *Random Forest* $Q_{95}$ regression model used known information about the watersheds in training set, namely: the watershed descriptors (independent variables) and the $Q_{95}$, in L/s/km$^2$, estimated from the observed historical series (dependent variable). The most relevant characteristics identified by *Random Forest* $Q_{95}$ method were: days of precipitation with monthly accumulation of 150 mm, basin centroid longitude, basin average slope, pasture fraction and forest fraction. $Q_{95}$ was calculated, for both observed and simulated flows, using calibration and validation periods. Thus, at least 15 years of fluviometric records were used to estimate the reference flow.

Correlations between observed $Q_{95}$ flows and those predicted by calibration and regionalization methods were calculated. Regionalizations with highest performances were obtained by physiographic-climatic similarity method, with correlation (R) of 0.973, and then *Random Forest* $Q_{95}$ method with a correlation of 0.965. $Q_{95}$ flows predicted by calibration of GR4J model had correlation of 0.9956, the regionalization method based on proximity had correlation of 0.9386, while *Random Forest* reached a correlation equal to 0.9392.

## 6 Conclusions

In this study, three regionalization methods were developed and deployed with the purpose of estimating daily flows in basins of Paraná State. A set of hydrometeorological data was created and presented together with catchments descriptive indexes. The amount of collected data is greater than national level datasets for the region, since a higher density of fluviometric stations were used. GR4J was employed for 126 watersheds and achieved optimistic performances in validation period (logNSE $\geq$ 0.70) for 65% of the watersheds.

All regionalization methods showed positive performances. Median values of logNSE in regionalization were equal to 0.702, 0.736 and 0.679 for spatial proximity, physiographic-climatic similarity and *Random Forest* methods, respectively. When comparing the median NSE between the three methods, *Random Forest* is slightly better. However, the median sqrtNSE was higher for the physiographic-climatic similarity method. The regionalization based on physiographic-climatic similarity proved to be the most robust method for predicting daily flow and $Q_{95}$ reference flow. When increasing the number of donor basins, the method based on spatial proximity has comparable performance to the method based on physiographic-climatic similarity.

Based on the physiographic-climatic characteristics of the basins, it was possible to classify 6 distinct groups of watersheds in Paraná. The basins showed similarities in their size, urban area fraction, average duration of consecutive dry days, number



of days with more than 150 mm of precipitation, and forest fraction. Interestingly, the last two descriptors were also relevant
for the *Random Forest* $Q_{95}$ model.

The low accumulated rainfall in years 2020 and 2021 caused a historic drought in Paraná. Due to the importance of water resources for supplying water to the population, irrigating crops and generating energy, it is expected that new research can be developed in the future based on this study and on information that were collected in this work. We recommend for future studies the use of stochastic optimization techniques for model calibration and the use of different hydrological models for
parameter regionalizations. In addition, we suggest the estimation of confidence intervals for the regionalized parameters and the use of regionalization methods based on geostatistical techniques.



# Appendix A: Data availability and physiographic/climatological indices

Figure A1 shows the availability of data over the years, the darker the shade of green the more data.. Watersheds descriptive characteristics are shown in Table A1, the region can be classified as humid subtropical climate (Matallo Junior, 2001).

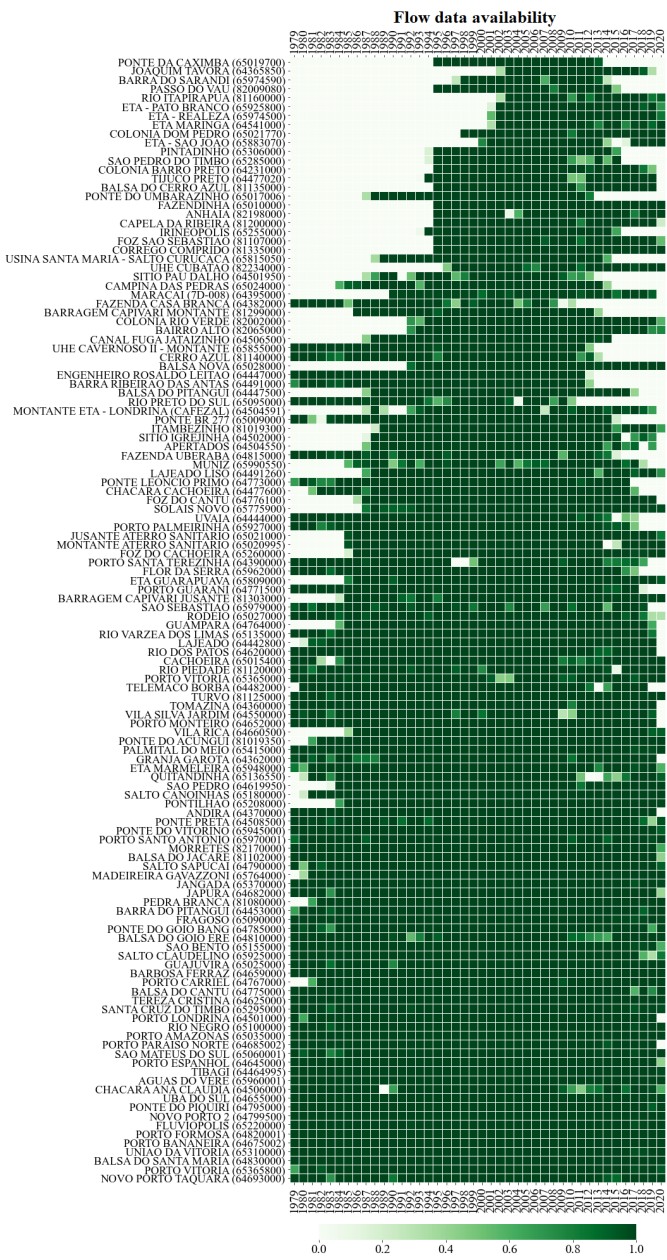

**Figure A1.** Availability of flow data by station. The darker the green color, the more data available for that year.





Table A1: Descriptive statistics for the Paraná dataset.

| Descriptors | mean | standard deviation | min | 25 % | 50 % | 75 % | max |
|---|---|---|---|---|---|---|---|
| **Physiographic indices** | | | | | | | |
| Mean altitude of the centroid (m) | 743.16 | 210.64 | 62.00 | 611.25 | 773.50 | 892.25 | 1132.00 |
| Area (km$^2$) | 4474.15 | 7001.22 | 13.87 | 510.70 | 1523.48 | 4120.90 | 34440.18 |
| Average height of the basin (m) | 804.33 | 171.96 | 262.85 | 666.36 | 835.74 | 920.45 | 1150.88 |
| Average slope of the basin (m/m) | 0.16 | 0.06 | 0.06 | 0.12 | 0.14 | 0.18 | 0.33 |
| Strahler number | 6.69 | 1.29 | 3.00 | 6.00 | 7.00 | 7.00 | 9.00 |
| Main river length (m) | 185335.43 | 164551.04 | 7336.06 | 66867.87 | 122924.13 | 236343.96 | 748033.70 |
| Drainage density (km/km$^2$) | 2.55 | 1.04 | 0.72 | 1.81 | 2.33 | 3.28 | 5.59 |
| Main river slope (m/m) | 0.04 | 0.02 | 0.02 | 0.03 | 0.03 | 0.04 | 0.10 |
| **Climatological indices** | | | | | | | |
| Coefficient of variation of annual precipitation | 0.17 | 0.02 | 0.13 | 0.16 | 0.17 | 0.18 | 0.21 |
| July average temperature (°C) | 15.55 | 0.58 | 14.56 | 15.16 | 15.42 | 15.98 | 17.02 |
| January average temperature (°C) | 22.96 | 0.46 | 22.26 | 22.56 | 22.89 | 23.31 | 24.07 |
| Precipitation days with monthly accumulation of 10mm | 152.86 | 17.45 | 112.79 | 142.02 | 151.56 | 162.55 | 213.14 |
| Precipitation days with monthly accumulation of 50mm | 146.07 | 17.55 | 104.38 | 133.40 | 145.44 | 156.40 | 208.45 |
| Precipitation days with monthly accumulation of 150mm | 82.84 | 17.43 | 51.38 | 70.54 | 81.52 | 93.96 | 159.12 |
| Annual potential evapotranspiration (mm) | 1255.95 | 78.02 | 1139.88 | 1192.19 | 1243.22 | 1326.41 | 1423.14 |
| Average annual precipitation (mm) | 1678.61 | 216.08 | 1357.26 | 1511.72 | 1614.79 | 1828.01 | 2618.98 |
| Average solar radiation in winter months (kwh/m$^2$) | 3.39 | 0.15 | 3.13 | 3.27 | 3.39 | 3.50 | 3.70 |
| Average solar radiation in summer months (kwh/m$^2$) | 5.53 | 0.19 | 5.16 | 5.37 | 5.54 | 5.71 | 5.86 |
| Aridity index | 1.34 | 0.18 | 0.97 | 1.24 | 1.30 | 1.45 | 2.02 |
| Average daily precipitation (mm/days) | 4.60 | 0.59 | 3.72 | 4.14 | 4.42 | 5.00 | 7.17 |
| Frequency of days without rain (days/year) | 208.28 | 17.94 | 148.24 | 197.65 | 209.87 | 219.21 | 249.17 |
| Average length of days without rain (days) | 4.54 | 0.43 | 3.41 | 4.29 | 4.57 | 4.78 | 5.71 |
| **Land use and land cover** | | | | | | | |
| (1) Forest (%) | 0.48 | 0.25 | 0.06 | 0.29 | 0.41 | 0.71 | 0.98 |
| (2) Agriculture (%) | 0.27 | 0.21 | 0.00 | 0.07 | 0.26 | 0.42 | 0.79 |
| (3) Urban area (%) | 0.03 | 0.07 | 0.00 | 0.00 | 0.01 | 0.02 | 0.53 |
| (4) Exposed soil (%) | 0.00 | 0.00 | 0.00 | 0.00 | 0.00 | 0.00 | 0.01 |
| (5) Pasture (%) | 0.23 | 0.13 | 0.01 | 0.14 | 0.19 | 0.29 | 0.80 |
| (6) Water (%) | 0.00 | 0.01 | 0.00 | 0.00 | 0.00 | 0.00 | 0.08 |
| Curve Number | 77.37 | 4.68 | 57.93 | 75.90 | 77.88 | 80.39 | 87.91 |
| **Soil type** | | | | | | | |
| (1) Latosol (%) | 0.24 | 0.18 | 0.00 | 0.08 | 0.25 | 0.35 | 0.78 |
| (2) Neosol (%) | 0.20 | 0.18 | 0.00 | 0.02 | 0.15 | 0.33 | 0.70 |
| (3) Argisol (%) | 0.14 | 0.17 | 0.00 | 0.00 | 0.11 | 0.21 | 0.91 |
| (4) Nitosol (%) | 0.10 | 0.13 | 0.00 | 0.00 | 0.04 | 0.15 | 0.65 |
| (5) Cambisol (%) | 0.20 | 0.23 | 0.00 | 0.02 | 0.11 | 0.31 | 0.95 |
| (6) Gleissolo (%) | 0.01 | 0.03 | 0.00 | 0.00 | 0.00 | 0.02 | 0.21 |
| (7) Organosol (%) | 0.01 | 0.03 | 0.00 | 0.00 | 0.00 | 0.00 | 0.18 |
| (8) Spodosol (%) | 0.00 | 0.00 | 0.00 | 0.00 | 0.00 | 0.00 | 0.00 |
| (9) Rocky outcrop (%) | 0.02 | 0.04 | 0.00 | 0.00 | 0.00 | 0.01 | 0.27 |
| (10) Urban Area (%) | 0.01 | 0.05 | 0.00 | 0.00 | 0.00 | 0.00 | 0.43 |



## Appendix B: Comparison of descriptors

Figure B1 shows the Pearson correlation coefficients between watershed descriptors.

**Figure B1.** Pearson Correlation Coefficients between descriptors.

*Competing interests.* The contact author has declared that none of the authors has any competing interests.

*Acknowledgements.* This study was partially financed by the Estonian Research Council (grant PRG 1674), and the European Union's Horizon 492 2020 Research and Innovation Programme (grant agreement no. 871115) ACTRIS IMP. The authors are grateful to Erasmus+ Eesti



Maaülikool (EMÜ) staff mobility program and to the Network on Environmental Monitoring and Modeling (RESMA) project from Federal University of Parana (UFPR) - Coordination for the Improvement of Higher Education Personnel (CAPES) - Institutional Internationalization Program (PRINT) for facilitating the exchange of researchers between Brazil and Estonia. This work was carried out with the support of Technology and Environmental Monitoring System of Paraná (Simepar) and the Sanitation Company of Paraná (Sanepar).



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
