# Peer review of "Regionalization of GR4J model parameters for river flow prediction in Paraná, Brazil"

_EGUsphere, 2023_

## Author Response (AR1)

Dear Editor Dr. Frederiek Sperna Weiland,

Here we present the authors response and article modifications after editor decision: Reconsider after major revisions on 17 Dec 2023 to egusphere-2023-1755.

We have improved the manuscript in line with the suggestions of reviewers, especially reviewer 2, according to the following:

1. The introduction was modified to describe precisely what the current research gaps are and how this study goes beyond the existing literature. We have highlighted the following advances:
   a. Understand regionalization techniques in subtropical climate.
   b. Proof-of-concept that basins without flow monitoring can have a good approximation of streamflow if other physiographic-climatic indices are provided.
   c. Machine learning algorithms perform better with physiographic-climatic indices as inputs.
2. The structure of the article was changed to Introduction -> Data for Paraná State -> Methods -> Results -> Discussion and -> Conclusions. This includes reorganizing the catchment descriptors into the Data section, put regionalization methods and GR4J model description into the Methods section and enlarge the Discussion with our novel contributions. We also improved the literature review in the introduction.
3. Equation (1) was corrected: the leading expression "1-" was missing on the right side.

All changes in the manuscript were highlighted with blue color. We have focused our improvements in revealing what of our research can be generalized to other basins.

Sincerely, the authors.

---

## Author Response (AR2)

Dear Editor Dr. Frederiek Sperna Weiland,

We are very grateful for all suggestions to improve the manuscript. Here we present the authors response and article modifications after editor decision: Publish subject to revisions on 19 Apr 2025 to egusphere-2023-1755. We have improved the manuscript in line with the suggestions of Report #1 by Referee #2.

The text colour in the manuscript was changed to blue to identify the modifications relating to the questions and points highlighted by reviewers. Our considerations are listed below, following the structure of the comments. The discussion received substantial improvement and we believe that this new version fulfils the reviewers and editor requirements. Please see specific responses below.

It is true that we have missed some interpretation of the findings related to hydrological processes. Our study reveals how land use modification influences the water availability for Paraná region and indicates some future possibilities for water resource management. Paraná has experienced severe draughts in recent years, affecting public distribution of water and the economy.

We have marked in orange the comments/questions/suggestions of Report #1 by Referee #2. Our response is in black colour in the following.

Dear authors,

I would like to thanks for considering of my initial comments. The comments related to the clarity of presentation and structure of the manuscript have been adequately addressed. What I still miss is the process orientation and interpretation of the findings. One of the main novelties of the research is to assess the regionalisation of the low flows in humid subtropical and hot temperate climates. It will be very important to indicate how the low flows are generated in such conditions, their seasonality and strength and how the selected regionalisation models (and clustered group) reflect these processes and patterns found.

Thanks for the comment, the following paragraphs were included in the manuscript (lines 390-420):

In dry periods, flow of rivers in Paraná are sustained basically by two mechanisms: baseflow and groundwater recharge. Even in periods with no precipitation, there can happen movement of water from underground aquifers and saturated soil layers into surface water bodies, such as rivers, lakes, or wetlands. All basins studied in this research are draining to the Paraná river, beneath which resides the Guarani aquifer, one of the largest sandstone aquifers in the world (Hirata and Foster, 2021). Karst terrains are also widespread throughout Paraná basin with the Açungui Karst and non-carbonate karsts being the most important ones (Auler and Farrant, 1996; Vestena and Kobiyama, 2007). Açungui carbonate karst is characterised by large areas of horizontally bedded limestones and dolomites, which form extensive regions of little or no relief, and are drained by low gradient rivers (Auler and Farrant, 1996). Interbasin groundwater flow may also play an important role in the water balance during dry periods in karst catchments (Vestena and Kobiyama, 2007).

Bartiko *et al.* (2019) identified that the rainfall season occurs in DJF months in the south of Brazil. The basins under study in Paraná State are in a region of climatic transition, with reasonably well-distributed rainfall throughout the year. Region's seasonality is generally divided between the six months centered around summer, from October to March, which correspond to the wet period, and the remaining months, from April to September, which correspond to the dry period. However, the occurrence of cold fronts, low-pressure areas, and instability systems during the Brazilian winter can provoke large floods even when this dry period and interrupt the recession process of the hydrographs.

In appendix C we included the hydrographs by physiographic-climatic similarity group. The comparison of hydrographs separated by groups of similar watersheds show the seasonality and strength of smaller flow rates. In these climatic conditions, the predominance of low flows is expected from April to September. The slow release of groundwater volumes after the cessation of surface runoff causes a recession curve that is strongly influenced by river-aquifer interaction. This curve, which conceptual models try to capture through simple mathematical relationships, is influenced by various factors, namely: soil properties, hydraulic characteristics and extent of aquifers, rate and amount of groundwater recharge, evaporation and evapotranspiration of the basin, spatial distribution of vegetation cover, among others (Musy et al., 2014).

Due to the complexity of hydrological processes and the specificities existing in the river basins, representing the recession curve and simulating low flows using conceptual models is an arduous process, sometimes requiring a basin-by-basin hydrological analysis. Attempts to improve this representation in the design of hydrological models have resulted in an increase in the number of parameters, as is the case with the traditional Sacramento Soil Moisture Accounting (SAC-SMA) model, which uses two conceptual reservoirs to simulate low flows with an overlay effect that allows better capture of low flow variability for a wider range of river basins (Burnash, 1995). The model applied in this study has an improved version, the GR6J, dedicated to low flows, which uses two additional parameters to better represent exchanges between the river and groundwater (Pushpalatha et al., 2011). In both cases, a better representation of low flows is achieved at the cost of increased model degrees of freedom, which is not ideal for regionalization issues.

For example, how and what is the impact of forest cover on low flows? Indicating and evaluating the log NSE efficiencies is fine, but some examples of how exactly the regionalised flows fit to observations will be interesting to see. The process interpretation of results can also be then better linked with previous studies in the Discussion section.

I saw your response that process interpretation goes beyond the scope of your analyses, but without it the significance of the results and new scientific contributions is very limited.

Thanks for the comment, the following paragraphs were included in the manuscript (lines 341-349):

Looking at parameters distributions (Figure 10) from a process perspective, we can find some relations with the catchment descriptors distributions (Figure 9).

Parameter x1 represents the runoff-producing capacity of the watershed reservoir. Our result shows that group 2, which has a higher percentage of forests, also have the highest x1

median and spread. This can support the hypothesis that more forest can improve the catchment capacity of generating runoff. Parameter x4 represents the base time of instantaneous unit hydrograph. The boxplots show that larger basins (group 4) have higher x4 parameters, i.e. bigger watershed areas may increase the base time of a hydrograph. Parameter x3 represents the propagation reservoir capacity. Our results show that group 6, which has more urban areas, also has the smaller x3 parameter. This reflects the effect of cities impermeabilization in terms of the flow propagation capacity, i.e. after a precipitation event the watersheds with more urban area have a smaller propagation capacity, or a fast response of the flow peak

Thanks for the comment. A paragraph was added to the conclusion (lines 466-472)

Our regionalization study showed that parameters are sensible to basin physiographic characteristics and soil use, and this have a direct effect on the streamflow response, i.e. hydrograph peak time, hydrograph base time, production capacity and propagation capacity. Urban impermeable areas produce a fast response of the flow peak. Forests play a significant role in groundwater recharge and low flow generation through various mechanisms: interception and slowing infiltration; enhancing soil structure and porosity and reducing erosion through root system soil stabilization. Overall, forests act as natural sponges, slowing down the movement of water, enhancing infiltration, and promoting groundwater recharge. Protecting and maintaining forest ecosystems is essential for sustaining groundwater resources and ensuring water availability for both human and natural systems.

Specific responses:

Abstract: "for transferring constants from hydrological models". The meaning of constants is unclear. Do you mean parameters?

Thanks for the correction in the Abstract: we rewrote the sentence "for transferring **constants** from hydrological models" and changed it to "for transferring **parameters** from hydrological models".

GRJ4 model description: It will be interesting and important to expand the model description from the runoff generation perspective and link it with processes responsible for low flows in the study region.

Thanks for the comment, as described before we added (lines 341-349) to the manuscript.

Figure 6,7. X axis labels. Why have values decimal numbers?

Thanks for the comment. We have corrected the X axis labels of Figures 6,7, now they don't have decimal numbers.

Figure 8,9,10. why group0? using zero might be confusing.

Thanks for the comment. We have renamed the groups of Figures 8,9,10, now the numbering starts in 1. The text numbering was also fixed.

Fig.10. what does it mean (from a process perspective) if in some group of catchments is e.g.the X4 parameter larger than in the other groups? Do the differences in parameter distributions between the groups make some difference in how low flows are generated? The results present that one group of catchments " have a higher percentage of forest as a distinct characteristic". Still it will be interesting to elaborate more about the implications of that on runoff and low flow generation. How is the larger forest relevant to runoff generation in these catchments?

Thanks for the comment. We have added a description of parameter's variation in a process perspective (lines 341-349).

Sincerely, the authors.

References:

Auler, A., & Farrant, A. R. (1996). A brief introduction to karst and caves in Brazil. Proceedings of the University of Bristol Spelaeological Society, 20(3), 187-200.

Bartiko, D., Oliveira, D. Y., Bonumá, N. B., & Chaffe, P. L. B. (2019). Spatial and seasonal patterns of flood change across Brazil. Hydrological Sciences Journal, 64(9), 1071-1079.

Burnash, R. J. C. (1995). The NWS River Forecast System-catchment modeling. Computer models of watershed hydrology., 311-366.

Hirata, R., & Foster, S. (2021). The Guarani Aquifer System–from regional reserves to local use. Quarterly Journal of Engineering Geology and Hydrogeology, 54(1), qjegh2020-091.

Vestena, L. R., & Kobiyama, M. (2007). Water balance in karst: case study of the Ribeirão da Onça catchment in Colombo City, Paraná State-Brazil. Brazilian Archives of Biology and Technology, 50, 905-912.

Musy, A., Hingray, B., & Picouet, C. (Eds.). (2014). Hydrology: a science for engineers. CRC press.

Pushpalatha, R., Perrin, C., Le Moine, N., Mathevet, T., & Andréassian, V. (2011). A downward structural sensitivity analysis of hydrological models to improve low-flow simulation. Journal of hydrology, 411(1-2), 66-76.